# Assessment of Hygiene Indicators When Using Gloves by Transport Workers in Russia during the COVID-19 Pandemic

**DOI:** 10.3390/ijerph19031198

**Published:** 2022-01-21

**Authors:** Elena Belova, Ekaterina Shashina, Yury Zhernov, Nadezhda Zabroda, Vitaly Sukhov, Olga Gruzdeva, Tatyana Khodykina, Evgenia Laponova, Valentina Makarova, Anton Simanovsky, Anastasia Zhukova, Tatiana Isiutina-Fedotkova, Denis Shcherbakov, Oleg Mitrokhin

**Affiliations:** 1Department of General Hygiene, F. Erismann Institute of Public Health, I.M. Sechenov First Moscow State Medical University (Sechenov University), 119435 Moscow, Russia; shashina_e_a@staff.sechenov.ru (E.S.); zhernov_yu_v@staff.sechenov.ru (Y.Z.); zabroda_n_n@staff.sechenov.ru (N.Z.); sukhov_v_a@staff.sechenov.ru (V.S.); khodykina_t_m@staff.sechenov.ru (T.K.); makarova_v_v@staff.sechenov.ru (V.M.); simanovskiy_a_a@staff.sechenov.ru (A.S.); isyutina-fedotkova_t_s@staff.sechenov.ru (T.I.-F.); shcherbakov_d_v@staff.sechenov.ru (D.S.); mitrokhin_o_v@staff.sechenov.ru (O.M.); 2Department of Chemistry, Lomonosov Moscow State University, 119991 Moscow, Russia; 3Department of Epidemiology, Federal State Budgetary Educational Institution of Further Professional Education «Russian Medical Academy of Continuous Professional Education» of the Ministry of Healthcare of the Russian Federation, 125993 Moscow, Russia; gruzdeva_oa@mail.ru; 4Department of Hygiene for Children and Adolescents, F. Erismann Institute of Public Health, I.M. Sechenov First Moscow State Medical University (Sechenov University), 119435 Moscow, Russia; laponova_ed@mail.ru; 5Department of Microbiology, Virology and Immunology Named after Academician A.A. Vorobiev, F. Erismann Institute of Public Health, I.M. Sechenov First Moscow State Medical University (Sechenov University), 119435 Moscow, Russia; zhukova_a_v1@student.sechenov.ru

**Keywords:** gloves, hygienic assessment, integral indicator, COVID-19, prevention

## Abstract

The COVID-19 outbreak represents a global public health emergency. National governments have gradually introduced restrictive measures. Using respiratory protective equipment (face masks) and gloves was essential practice without specific infection control measures or guidelines. This study aimed to assess hygiene indicators when using gloves by transport workers in Russia during the COVID-19 pandemic and to develop integral indicators and recommendations for wearing gloves for workers for whom this is a mandatory requirement. For this purpose, 1103 transport workers were surveyed using a questionnaire. We investigated the hygiene aspects of gloves and evaluated the bacterial contamination of the contact side of the gloves based on the wash results. We assessed the socio-demographic characteristics of the respondents, the duration of the work shift, the frequency of use of types of gloves, skin manifestations, the degree of comfort, and bacterial growth. We carried out the ranking according to comfort, the absence of adverse dermatological reactions when wearing gloves, and bacterial contamination of the inner surface of the gloves. It has been identified that it is necessary to use a comprehensive assessment of gloves to create a register of protective equipment, taking into account the frequency with which it was worn, the severity of the skin condition, comfort, and bacterial contamination.

## 1. Introduction

The COVID-19 pandemic has led to the emergence of restrictive measures worldwide. In the absence of controlling infection-specific methods, priority was given to the requirements for the mandatory use of nonspecific protective equipment, including the use of personal protective equipment for the respiratory organs (face masks) and hands (gloves) [1].

It should be noted that the World Health Organization (WHO) does not have recommendations for wearing gloves in public places for non-medical professionals. There are only established policies for medical workers that come into contact with patients [2,3]. The International Labor Organization (ILO) proposes gloves for public works programs [4].

The US Centers for Disease Control and Prevention (CDC) points out the need to use gloves in the general public only when caring for sick people [5]. The CDC does not recommend wearing gloves in public and non-medical workplaces [6]. However, it notes that surface disinfection effectively prevents SARS-CoV-2 from secondary transmission between an infected person and others in homes and compliance with hand hygiene and surface treatment of premises [7].

The Occupational Safety and Health Administration of the US Department of Labor (OSHA) suggests the need for hand protection for non-medical workers of medium exposure risk (workers in crowded areas, educational institutions, retail organizations, etc.) [8].

The European Center for Disease Prevention and Control (ECDC) also has no recommendation for wearing gloves in public and for non-medical workers [9]. Despite this, some European Union countries have introduced requirements for employers to provide employees with hand protection [10,11].

In Asian countries, the requirements for wearing gloves vary by country. The China Center for Disease Control and Prevention (China CDC) recommends wearing gloves when flying in an airplane [12]. The WHO mission in China, in its report, does not recommend the use of hand protection in the country by anyone other than health workers [13]. The Government of the Republic of Korea obliges public transport employees to wear gloves and recommends that others wear hand protection when visiting shops and supermarkets. Mass events in the Republic of Korea are held with strict guidelines with regard to the use of masks and gloves. [14]

In the Russian Federation, since February 2020, measures (administrative, legislative, organizational, technical, and hygienic) have been gradually introduced to prevent the emergence and spread of COVID-19 [15,16].

In May 2020, the Russian Federation introduced a requirement for the population to wear respiratory (masks, respirators) and hand (gloves) protection while in transport [17], which was later expanded to include visits to public facilities [18].

The requirement to wear gloves remained throughout the country even after the restrictions were partially lifted [19,20]. Due to mass vaccinations and a decrease in the number of new cases of COVID-19 in some regions of Russia, the mandatory wearing of gloves was canceled [21]. However, the need for the compulsory wearing of gloves persists for persons of certain professions associated with the risk of infection with COVID-19 [22]. Gloves are believed to prevent contamination of workers’ hands and protect the wearer from infectious agents, and reduce the risk of contamination by contact by preventing contaminated hands from touching the face [23,24]. Case reports indicate that SARS-CoV-2 is transmitted between people by touching surfaces that the patient has recently coughed or sneezed on and then touching one’s mouth, nose, or eyes [25,26]. However, no document specifies what kind of hand protection is recommended for people from different non-medical professions.

The recommendations in force in Russia list the possible use of hand protection for broad professional groups without justification and taking into account the specifics of various professions. Thus, all transport workers are advised to use gloves made of polymeric materials (medical diagnostic gloves, household rubber, or polyethylene gloves) [22].

Consideration should be given to the available data on the prevalence of adverse skin reactions caused by gloves, including contact dermatitis, erythema, xerosis, pruritus, desquamation, and/or hyperpigmentation of the skin of the hands [27,28,29]. Wearing gloves in the workplace affects the prevalence of occupational eczema. This is true for medical [30,31] and non-medical workers [32]. The majority of occupational eczema in health care workers increased significantly during the pandemic [33].

Many adverse skin reactions have been reported with different types of gloves, including irritant contact dermatitis, allergic contact dermatitis, and contact urticaria [27,34]. The most common irritant, contact dermatitis, causes skin to become dry, with crusted patches and fissures [35]. When questioning medical workers, 65.9% of respondents reported skin damage due to prolonged wearing of gloves (more than 6 h) [36]. According to Chinese researchers, 88.5% of users of latex gloves develop skin reactions such as local itching, burning, tingling, and contact and generalized urticaria [37].

The existing variety of hand protection products offered on the market, most of which do not have a registration certificate from an authorized state body, the lack of uniform approaches to the labeling of manufactured protective equipment, and clear recommendations to the population for their choice determines the need to systematize the available information and develop a unified methodology for the hygienic assessment using the influence of different types of protective equipment under other working conditions.

This study aimed to assess hygiene indicators of transport workers when using gloves in Russia during the COVID-19 pandemic and to develop indicators and recommendations for wearing gloves for workers for whom this is mandatory.

## 2. Materials and Methods

### 2.1. Questionnaire

A survey of transport service employees was conducted during the COVID-19 pandemic, when wearing gloves was mandatory. The research was conducted in August 2021.

The employees of the transport service took part in the survey. The questionnaire was developed by the staff of the Sechenov University General Hygiene Department. The working conditions, the type of gloves used by workers of different professions, reactions after wearing gloves in frequency and severity, and the degree of comfort in using different types of gloves were assessed. The total number of involved participants (respondents) was1103. In total, 1103 questionnaires were completed and submitted for processing, of which 928 questionnaires were admitted for statistical processing after a thorough assessment of the quality of the responses.

The inclusion criteria included adult employees working in various positions in the transport industry that had no medical restrictions with regard to working for the period of the study, and the carrying out of professional activities in the workplace with gloves during at least one shift (12 h). All participants voluntarily agreed to participate in the study.

### 2.2. Types of Gloves

By analyzing the range of personal protective equipment used to cover the skin of the hands and the study of the chemical composition of the material from which the gloves is made, it possible to combine gloves into five groups (Table 1). An additional criterion for the formation of groups, in our opinion, can be determined by the consumer demand of transport workers for this type of gloves. 

This study excluded commercial interests when choosing personal protective equipment for the hands (gloves). The choice of gloves for the study was carried out without considering the brand or the recommendations of glove manufacturers.

### 2.3. Bacterial Contamination

The study of the bacterial contamination of the gloves was carried out immediately after they were worn. The swabs were taken from the inner contact surface of the used gloves.

The types of gloves that participated in the study included the following types: (a) synthetic thin elastic (disposable; nitrile and vinyl), (b) polyethylene, thin (disposable); (c) fabric, cotton, knitted; (d) those combined with a polymer coating. Washes were inoculated onto nutrient media of GRM-agar. The samples were incubated in a thermostat at 30 ± 1 °C. Then the growing colonies of microorganisms were counted. We recorded a number of colony-forming units (CFU/cm^3^) grown on each type of glove worn during different time intervals (2 and 12 h). The study was conducted in the warm season at an ambient temperature of 24 ± 1 °C.

The study did not evaluate the viral permeability of personal protective equipment for hands due to the peculiarity of the structure of the glove material used by transport workers in conditions of mandatory wear during the COVID-19 pandemic.

### 2.4. Statistical Analysis

Research results were analyzed and processed using the statistical software package STATISTICA Base (TIBCO Software Inc., Palo Alto, CA, USA). Data analysis included the calculation of absolute mean values (M ± SD) and relative values. Extensive values were presented as a percentage. We determined the significance of the differences in features based on the value of the Pearson fit criterion (χ^2^). A statistical study of the relationship between the features was carried out using Spearman’s nonparametric correlation coefficient (r) with Fisher’s transformation (z) to approximate the exact distribution of the correlation coefficient. The comparison of glove types and severity of bacterial growth after wearing for 2 and 12 h was carried out using the nonparametric Kruskal-Wallis test. The null hypothesis was that there was no difference between the groups. The critical value of the significance level (*p*-value) of the testing hypotheses was calculated as *p* ≤ 0.01.

## 3. Results

### 3.1. Socio-Demographic Characteristics of Respondents

The main socio-demographic characteristics of the respondents are presented in Table 2.

The absence of statistically significant age differences between the employees who took part in our study (divided by gender) allows us to carry out calculations for the combined group of respondents in the future.

### 3.2. Shift Length

Analysis of the respondents’ answers to the question about the duration of the work shift showed that for 63.15% of the respondents (*n* = 586), it was between 8–12 h; the duration of a change of 8 h was for 24.46% of the respondents (*n* = 227), over 12 h-for 12.39% (*n* = 115).

### 3.3. Frequency of Gloves Types Use by Workers

The most frequently used gloves by respondents were the following types: synthetic thin elastic (disposable; nitrile and vinyl) 44.29% (*n* = 411), fabric, cotton, knitted-25.75% (*n* = 239), polyethylene, thin (disposable)-18.43% (*n* = 171), combined with a polymer coating-11.53% (*n* = 107).

The choice of gloves by a particular employee depends on the nature of the work performed (work process) and the duration of wearing gloves during the work shift (r 0.29–0.48, *p* < 0.01).

### 3.4. Skin Manifestations and Comfort Degree

The respondents assessed the frequency and severity of local skin issues and the degree of comfort when wearing gloves. For all types of reactions, a strong correlation was found between the frequency of wear and the severity of the corresponding responses, in general, the more often the reaction was manifested, the stronger was the degree of its severity (r 0.88–0.91; *p* < 0.01).

The analysis showed that the most frequent and pronounced strong reactions to wearing gloves among the respondents were: sweating of the hands, which was observed in 57.49% of the respondents (χ^2^ = 116.401; *p* < 0.001), while redness, peeling, and irritation of the skin was noted by only 25.34% (χ^2^ = 56.630; *p* < 0.001); least of all, respondents reported such reactions to wearing gloves as pimples, rash, inflammation, and cracks 7.64% (χ^2^ = 14.733; *p* < 0.001). The respondents who indicated the presence of an acute or chronic skin disease noted the appearance of pustules, rash, inflammation, and cracks when wearing gloves (r 0.09; *p* < 0.01).

The frequency and severity of the reactions were inversely proportional to the degree of comfort when wearing gloves (r −0.21–−0.34; *p* < 0.01). The results of comparing different types of gloves in terms of the frequency and severity of subjective reactions are presented in Table 3.

In addition to calculating the Kruskal-Wallis *H* test and its significance level, we ranked the average respondents’ assessments in terms of frequency, the severity of skin reactions, and the comfort of use for different types of gloves. The sum of the ranks made it possible to single out the gloves with the most and most minor pronounced skin reactions. The respondents’ assessment of the lack of adverse skin reactions and the comfort from the use of each type of gloves increases from least to most skin-friendly/comfortable in the following order: combined with a polymer coating (10 points)—fabric, cotton, knitted (15 points)—synthetic thin elastic (disposable; nitrile and vinyl) (20 points)—plastic, thin (disposable) (25 points).

### 3.5. Bacterial Growth

The ratios compared to the control values of bacterial contamination of the inner surface of gloves after wearing by respondents for 2 and 12 h are presented in Figure 1.

The statistical hypothesis about the existing differences in the dynamics of the growth of colony-forming units on the material of nitrile and vinyl gloves was not confirmed in the course of the study. Therefore, we referred gloves made of these materials to the general group for an integrated assessment.

The questionnaire analysis showed that the respondents’ wearing of thin polyethylene (disposable) gloves led to the most significant discomfort and the appearance of skin reactions. This type of glove was excluded from the bacteriological study due to the respondents’ high degree of irritation during wear.

Statistically significant differences were found in all studied groups of gloves in terms of the duration of wearing in relation to the control (*p* < 0.01). The intensity of colony growth on fabric cotton knitted gloves after 2 h and after 12 h of wearing was minimal (*p* < 0.01).

### 3.6. Ranking

The ranking was carried out according to comfort, absence of side dermatological reactions when wearing gloves, and bacterial contamination of the inner surface of gloves. Each of the characteristics we selected was evaluated separately. The best value for each characteristic was awarded the highest score, based on the number of types of gloves analyzed. Ranks for assessing comfort and CFU: highest score-3, lowest-1; ranks for assessing skin reactions: the highest score is 1, the lowest is 3. The sum of the ranks gives an integral assessment of the gloves (Table 4). The higher the overall hygienic score of the glove, the more comfortable and safe the glove is to use. Based on the correlation analysis, a reduction factor of 0.1 was calculated for the questionnaire survey results (assessment of the frequency and severity of skin reactions and wearing comfort).
*CR* = ((*Comf*. + *DR*) × 0.1),


*where I = CR + CFU,*



*I—integrated assessment of PPE*



*CR—gloves comfort rating*



*DR—dermatological reactions*



*Comf.—comfort in wearing PPE*


## 4. Discussion

The Russian government, during the COVID-19 pandemic, introduced requirements for the wearing of personal protective equipment for the skin of hands (gloves) for all segments of the population on when visiting public places and using public transport. After removing the requirements for wearing PPE on the hands, the wearing of gloves at workplaces continued in the transportation sector [38].

Analysis of skin reactions to wearing different types of gloves was carried out during labor-intentional activity. The choice of the type of gloves by the respondent was conditioned by the specifics of the performed labor functions.

According to the data obtained, the most frequent and pronounced effect noted by workers when wearing gloves was sweating of the skin of the hands (57.49%; χ^2^ = 116.401; *p* < 0.001). Other skin reactions noted by the workers were redness, peeling, and skin irritation (25.34%; χ^2^ = 56.630; *p* < 0.001). Such manifestations can be associated with insufficient air permeability of the material from which the gloves are made,; the development of excess moisture created under the glove, which leads to skin maceration; erosion and disruption of the epidermal barrier [27,34]; and allergic reactions to chemical components of protective equipment (a mixture of thiuram and tetraethylthiuram disulfide; preservatives-formaldehyde and isothiazolinones [39]; tricresyl phosphate [40]; the use of nitrile and vinyl gloves is associated with an allergy to rubber accelerators, which leads to itching and redness [41,42]).

The listed reactions cause discomfort when wearing gloves and can lead to a weakening of attention and a decrease in concentration when performing professional tasks, which can cause industrial injuries. These reactions can also pose a risk to users’ health and lead to skin diseases. Thus, a hygienic assessment of the gloves used and the identifying of the causes of adverse reactions during their long-term use is necessary [43].

The study results of bacterial contamination show a statistically significant increase in CFU after 2 h of wearing gloves in relation to the control (*p* < 0.01). Statistically significant differences in the dynamics of wearing from 2 h to 12 h were found only in the fabric group that wore cotton, knitted gloves (*p* < 0.01). There were no statistically significant differences between wearing for 2 h and 12 h (*p* > 0.05).

## 5. Conclusions

The revealed hygienic indicators of gloves used during the COVID-19 pandemic by employees employed at transport facilities in Russia made it possible to identify the main components for formulating an integral indicator. The calculation of the integral indicator (from the maximum to the minimum sum of points) made it possible to place the gloves in the following order of preference: (1) synthetic, thin, elastic (disposable)–3.0; (2) fabric, cotton, knitwear–4.4; (3) combined with a polymer coating–2.9.

The statistically significant growth of bacterial colonies, compared with the control, was recorded by us in as little as 2 h of wearing all types of gloves studied, all the dependences of the material and structure of gloves. It can serve as the basis for wearing gloves by one worker for no more than 2 h.

The importance of studying the use of gloves by transport workers during the spread of COVID-19 is significant not only because of the importance of transport infrastructure and keeping it in working order, but also because it can serve as a model for planning a set of preventive measures and apply them to other areas of activity.

This integral assessment can be used in the hygienic analysis of types of gloves not included in this study. We recommend that the study of bacterial contamination of the inner surface of the glove be particularly noted as an important hygienic indicator. 

The accumulation of data and subsequent analysis using the proposed integral assessment will serve to create a register of protective equipment that takes into account the frequency and severity of skin reactions, comfort, and bacterial contamination. This will allow non-medical workers to make educated choices in terms of hand protection.

## Figures and Tables

**Figure 1 ijerph-19-01198-f001:**
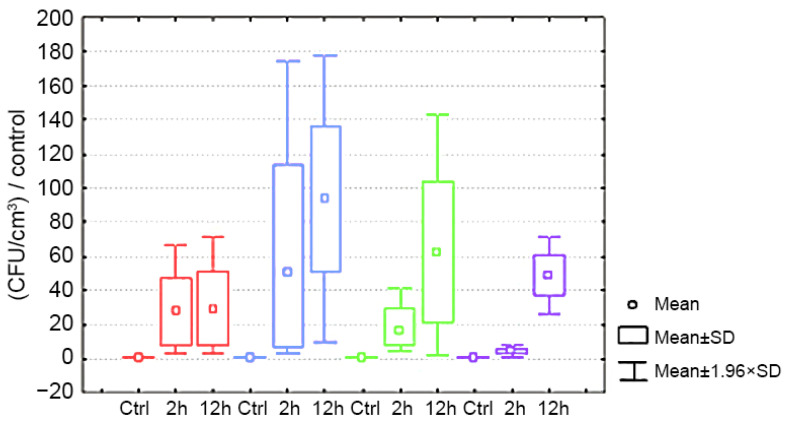
Bacterial contamination of different types of gloves depending on the time of wearing. The red box plots show the synthetic thin elastic disposable nitrile gloves. The blue box plots show the synthetic thin elastic disposable vinyl gloves. The green box plots show the combined gloves with a polymer coating. The violet box plots show the fabric, cotton, knitted gloves. Ctrl—control; 2 h—after 2 h of wearing; 12 h—after 12 h of wearing; CFU—colony-forming units.

**Table 1 ijerph-19-01198-t001:** Main characteristics of gloves included in the study.

Types of Gloves	Chemical Composition	Additional Characteristics
Synthetic, thin, elastic (disposable) 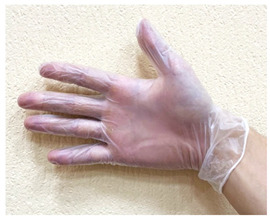	Vinyl	Examination gloves (non-sterile and sterile), sterile surgical gloves
Synthetic, thin, elastic (disposable) 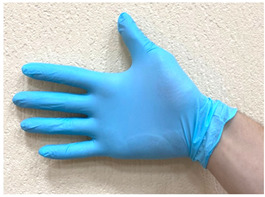	Nitrile	Examination gloves (non-sterile and sterile), sterile surgical gloves
Polyethylene, thin (disposable) 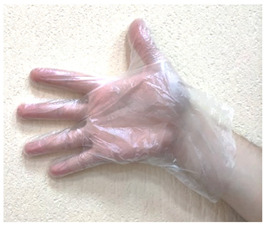	Polyvinylchloride	
Fabric, cotton, knitted 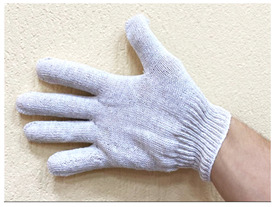	Cotton thread, wool thread, acrylic thread	Cotton gloves
Combined with a polymer coating 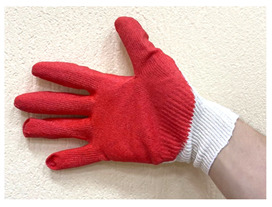	Cotton thread, wool thread, acrylic thread, natural latex doused on the contact surface	Cotton gloves, different grades

**Table 2 ijerph-19-01198-t002:** Socio-demographic characteristics of respondents.

Characteristic	Total (*n* = 928)	Men (*n* = 647)	Women (*n* = 281)
Age M ± SD	40.80 ± 10.80	39.62 ± 10.91	41.60 ± 11.19
min–max	19–71	20–70	19–71

**Table 3 ijerph-19-01198-t003:** Comparison of glove types in terms of frequency, the severity of subjective reactions, and comfort of use.

Characteristics	Type of Gloves								Kruskal–Wallis *H* test
	Synthetics *n* = 411		Polyethylene *n* = 171		Cotton *n* = 239		Combined *n* = 107		
	Means	Rank	Means	Rank	Means	Rank	Means	Rank	
Frequent sweaty hands	3.54	4	3.46	3	2.23	2	2.21	1	<0.001
redness, flaking, skin irritation	1.75	3	1.9	4	0.8	2	0.57	1	<0.001
pustules, rash, inflammation, cracks	0.67	3	0.97	4	0.39	2	0.1	1	<0.001
Severely sweaty hands	3.34	3	3.4	4	2.29	2	2.07	1	<0.001
redness, flaking, skin irritation	1.54	3	1.9	4	0.92	2	0.51	1	<0.001
pustules, rash, inflammation, cracks	0.53	3	0.95	4	0.51	2	0.17	1	<0.001
The comfort of glove use	2.17	1	2.58	2	2.99	3	3.06	4	<0.001
TOTAL		20		25		15		10	

**Table 4 ijerph-19-01198-t004:** Ranking of gloves by comfort, absence of side dermatological reactions in workers, and bacterial contamination of the inner surface.

Type of Gloves	Comf.	DR	CFU	Sum
synthetic, thin, elastic (disposable)	1	19	1	3.0
fabric, cotton, knitted	2	12	3	4.4
combined with a polymer coating	3	6	2	2.9

## Data Availability

The data presented in this study are available on request from the corresponding author.

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
