# Peer review of "Assessment of Hygiene Indicators When Using Gloves by Transport Workers in Russia during the COVID-19 Pandemic"

_ijerph, 2022, doi:10.3390/ijerph19031198_

Round 1

Reviewer 1 Report

The discussion is too short and it is little argued. 

The topic is not too original or relevant. It does not provide new information compared with other publications. The methodology is not very appropriate and the statistic have to be improved. The discussion and conclusions have to be better elaborated and justified.

Author Response

Thank you very much for your fair comments regarding our manuscript. Wearing gloves by all segments of the population during the difficult period of the spread of coronavirus infection was one of the elements of restrictive measures in the Russian Federation. After removing the requirements for the population to wear PPE for the skin of the hands, the practice of wearing gloves in the workplace continued in the field of transport. New data were introduced into the manuscript's text in the ‘Discussion’ section, and the text of the ‘Conclusions’ section was also finalized.

Reviewer 2 Report

The study is interesting but from a more commercial than a scientific point of view. The introduction refers in many places to COVID-19, but materials, methods, and results do not mention this relationship. On the other hand, the way to obtain the results is very basic, and the results do not have a relevant scientific value. With improvements in a larger microbiological analysis that include viral contamination the work could be considered for short communication. For these reasons, in my opinion, the present work should be rejected for publication.

Author Response

Reviewer 2: 

The study is interesting but from a more commercial than a scientific point of view.

Thank you for your comment. The article's authors exclude commercial interest when choosing personal protective equipment for the skin of hands (gloves). The choice of gloves for the study was carried out without considering the brand and recommendations of glove manufacturers.

This explanation has been included in the text of the article.

The introduction refers in many places to COVID-19, but materials, methods, and results do not mention this relationship.

Thank you very much for the precise remark in this section. New data were introduced in each section confirming this relationship.

On the other hand, the way to obtain the results is very basic, and the results do not have a relevant scientific value. With improvements in a larger microbiological analysis that include viral contamination the work could be considered for short communication. For these reasons, in my opinion, the present work should be rejected for publication.

The study did not assess the viral (SARS-CoV-2) permeability of personal protective equipment for hands due to the structure of the glove material used by transport workers in conditions of forced wear during the COVID-19 pandemic.

Reviewer 3 Report

Dear authors I would suggest the following :

45 replace varies for vary 

64-66 the period needs to be reformulated more clearly 

I think your paper requires moderate english changes and the conclusions would benefit from a clearer explanation of the importance of the results achieved in relation to what written in the introduction. 

1. The main question addressed by the research is to conduct an hygienic assessment of the gloves by workers employed at transport but it should be a systematization of the available information and should  develop an unified methodology for the hygienic assessment using the influence of different types of protective equipment.
2. I don't consider the topic of this manuscript original or relevant in the field , because there have been a lot of reports of occupationally induced skin conditions in healthcare workers related to the use of personal protective equipment.
3. It adds the comparison between different types of gloves in relation to the performed labor function compared with other published material.
4. I think that the authors did not clearly indicate  the objectives of the work.
5.The conclusions aren't consistent in relation to the main question of the research, rather it is  a list of recommendations.
6.The references are appropriate.
7.Statistical analysis is correct , but the statistical hypotesis about the existing differences in the dynamics of the growth of UFC on different gloves was not confirmed in the study. 

Author Response

Reviewer 3: 

Dear authors I would suggest the following:

45 replace varies for vary

Thank you very much for your valuable comments on the text of the manuscript. For us it is important. The remark has been corrected.

64-66 the period needs to be reformulated more clearly

The remark has been corrected.

I think your paper requires moderate english changes and the conclusions would benefit from a clearer explanation of the importance of the results achieved in relation to what written in the introduction.

Thank you, we paid attention to the quality of the English language, some changes were made. These sections have undergone changes and have been filled with information.

1. The main question addressed by the research is to conduct an hygienic assessment of the gloves by workers employed at transport but it should be a systematization of the available information and should  develop an unified methodology for the hygienic assessment using the influence of different types of protective equipment.

The conducted hygienic indicators of gloves used during the COVID-19 pandemic by employees employed at transport facilities in Russia made it possible to identify the main components for formulating an integral indicator. The calculation of the integral indicator (from the maximum to the minimum score) made it possible to place the gloves in the following sequence:

- synthetic, thin, elastic (disposable) - 3.0;

- fabric, cotton, knitwear - 4.4;

- combined with a polymer coating - 2.9.

This explanation has been included in the text of the article.

2. I don't consider the topic of this manuscript original or relevant in the field, because there have been a lot of reports of occupationally induced skin conditions in healthcare workers related to the use of personal protective equipment.

Wearing gloves by all segments of the population during the difficult period of the spread of coronavirus infection in the Russian Federation was one of the elements of restrictive measures. After the removal of the requirements for the population to wear PPE for the skin of the hands, the practice of wearing gloves in the workplace continued in the field of transport.

3. It adds the comparison between different types of gloves in relation to the performed labor function compared with other published material.

Thank you. In the Introduction, information has been added on the occurrence of skin reactions when wearing different types of gloves during the work of medical workers.

4. I think that the authors did not clearly indicate  the objectives of the work.

The purpose of the work was reformulated and edited.

5.The conclusions aren't consistent in relation to the main question of the research, rather it is  a list of recommendations.

The Conclusions has also undergone changes, as required in your fair remark.

6.The references are appropriate.

Links are correct, requires no change.

7. Statistical analysis is correct, but the statistical hypothesis about the existing differences in the dynamics of the growth of UFC on different gloves was not confirmed in the study.

Thank you. The statistical analysis performed didn’t reveal differences in the growth of UFC on different gloves in the study, but a dependence trend was shown, which requires further research.

This explanation has been included in the text of the article.

Round 2

Reviewer 1 Report

No comment.

Reviewer 2 Report

The article has been modified, you should accept its publication